# The ATP-Binding Cassette (ABC) Transport Systems in *Mycobacterium tuberculosis*: Structure, Function, and Possible Targets for Therapeutics

**DOI:** 10.3390/biology9120443

**Published:** 2020-12-04

**Authors:** Marcelo Cassio Barreto de Oliveira, Andrea Balan

**Affiliations:** Laboratório de Biologia Estrutural Aplicada, Department of Microbiology, Institute of Biomedical Sciences, University of São Paulo, São Paulo 05508000, Brazil; mbarretoliveira@hotmail.com

**Keywords:** *Mycobacterium tuberculosis*, ABC transporters, structure, drug-efflux, importers, mce proteins

## Abstract

**Simple Summary:**

*Mycobacterium tuberculosis* is a bacterium of great medical importance because it causes tuberculosis, a disease that affects millions of people worldwide. Two important features are related to this bacterium: its ability to infect and survive inside the host, minimizing the immune response, and the burden of clinical isolates that are highly resistant to antibiotics treatment. These two phenomena are directly affected by cell envelope proteins, such as proteins from the ATP-Binding Cassette (ABC transporters) superfamily. In this review, we have compiled information on all the *M. tuberculosis* ABC transporters described so far, both from a functional and structural point of view, and show their relevance for the bacillus and the potential targets for studies aiming to control the microorganism and structural features.

**Abstract:**

*Mycobacterium tuberculosis* is the etiological agent of tuberculosis (TB), a disease that affects millions of people in the world and that is associated with several human diseases. The bacillus is highly adapted to infect and survive inside the host, mainly because of its cellular envelope plasticity, which can be modulated to adapt to an unfriendly host environment; to manipulate the host immune response; and to resist therapeutic treatment, increasing in this way the drug resistance of TB. The superfamily of ATP-Binding Cassette (ABC) transporters are integral membrane proteins that include both importers and exporters. Both types share a similar structural organization, yet only importers have a periplasmic substrate-binding domain, which is essential for substrate uptake and transport. ABC transporter-type importers play an important role in the bacillus physiology through the transport of several substrates that will interfere with nutrition, pathogenesis, and virulence. Equally relevant, exporters have been involved in cell detoxification, nutrient recycling, and antibiotics and drug efflux, largely affecting the survival and development of multiple drug-resistant strains. Here, we review known ABC transporters from *M. tuberculosis*, with particular focus on the diversity of their structural features and relevance in infection and drug resistance.

## 1. Introduction

*M. tuberculosis* is the agent responsible for tuberculosis, with about 10 million people infected in the world in 2018, of which 1.2 million died in 2019 [1]. To make this scenario worse, many individuals have latent infections that are reactivated at some point in their lives, especially among those with other immunodeficient diseases such as co-infection with the HIV virus (human immunodeficiency virus). The bacillus has a cytoplasmic membrane, cell wall, and an external capsule formed mainly by polysaccharides that play a relevant role in the virulence of *M. tuberculosis* strains. The inner membrane is formed by a phospholipid bilayer with several integral membrane proteins, among which the important family of ATP-dependent transporters, the ATP-Binding Cassette (ABC) transporters, are also present. The cell wall or mycomembrane is formed by a thin layer of peptidoglycan associated with lipoarabinomananan and a layer of arabinogalactones. In addition, there are mycolic acids and other important components specific to the mycobacteria, such as different types of phosphosugars, trehalose, glycerol sulfolipids, and phenolic glycolipids [2].

Transport across the cell membrane is extremely important for the bacterium. Different molecules and substrates are required for diverse cellular processes. Species from the *M. tuberculosis* complex (MTBC) and strains from pathogenic groups have around 160 to 200 proteins dedicated to the ATP-dependent transport systems according to the TransportDB webserver [3] indicating the relevance of this protein family for the genus. In *M. tuberculosis*, 2.5% of the genome encodes components of ABC transporters, indicating their critical role in the maintenance of cellular homeostasis. These transporters, in addition to being responsible for the transport of important molecules such as sugars, amino acids, and many others, also act as efflux pumps of external agents and are thereby responsible for resistance to antibiotics [4]. Structurally, ABC transporters consist of a canonical set of two transmembrane domains (TMDs) that form a pore for the passage of the substrate (TMDs) and two nucleotide-binding domains (NDBs) that provide energy for substrate translocation through ATP hydrolysis. These components can be translated into independent polypeptides or joined in various possible combinations, forming homodimers or heterodimers [5]. In addition, ABC import systems have a third component, a periplasmic substrate-binding protein (SBP), which is responsible for the uptake and delivery of the substrate into the pore formed by the TMDs [6]. Based on the layout and architecture of the TMDs, ABC transporters can be divided into seven families, three importer families, and four exporter families [7]. Using bioinformatics analysis, structural characterization, and data mining, we have classified *M. tuberculosis* ABC transporter components into aforementioned families (Figure 1). 

Most ABC importer systems that have been identified in *M. tuberculosis* so far belong to group I and include systems dedicated to the import of sugars, amino acids, peptides, and anions. However, four systems related to iron import have been classified into group II and a unique cobalt transporter into group III (Energy Coupling Factor, ECF). 

*M. tuberculosis* ABC exporter systems fall into groups IV, V, and VII. It is interesting to observe that the secondary and or tertiary structural characteristics of some components do not fit their functional groups, as shown by experimental analysis. These components are IrtAB, BacA, Rv2563/64, Rv0072/73, Rv1747, and Rv0987/86. Functionally, IrtAB and BacA are importers, but their structures are very similar to the structures of ABC exporters. No periplasmic components were identified for these systems in the vicinity of the genome, and IrtAB also has an additional cytoplasmic domain which is thought to function in exportation. Similarly, Rv2563/64 and Rv0072/73 are structurally related to type VI transporters, although they have been associated with the import of amino acids due the presence of the NBD component. Rv2564 and Rv0073 are highly similar to the *E. coli* GlnQ protein, the ATPase subunit from the glutamine transporter [9]. Rv1747 and Rv0987/86, putative exporters, have a non-canonical domain architecture, with two cytoplasmic phosphorylated Fork-Head Associated (FHA) domains and two extracellular domains per TMD, respectively. This mixing of functions and structural features across families highlights how surprising and captivating the components of the *M. tuberculosis* inner membrane are and how little it is known about them and the proteins that belong to the bacillus envelope. Finally, we have the Mce proteins, whose function is critical for the bacillus existence. Nevertheless, their structural organization is not clear and only based on homology of the MceA-F components with the *E. coli* Mla proteins [10,11]. 

In this review, we summarized the current understanding of the putative ABC transporters in *M. tuberculosis*, their structural properties, and their functional role in the bacillus. We highlight the fact that the number of transporters in *M. tuberculosis* must be much higher than described, since searches for orthologues are hampered by the low sequence identity between these proteins and their poor functional characterization. 

## 2. Identified ABC transporters in *Mycobacterium tuberculosis*


### 2.1. ABC Transporters Type Importer

ABC importers are present in prokaryotes but not in eukaryotes. They are responsible for the import of different substrates including oligopeptides, amino acids, inorganic anions, sugars, and metals. Because of such a diverse range of substrates, ABC importers are involved in many physiological processes, including transport, nutrition, virulence, and pathogenesis. Most ABC importers in *M. tuberculosis* are responsible for the bacillus adaptation in the human host. In this section, we describe the components and full ABC importers systems that were identified in *M. tuberculosis*, including their roles to keep the bacillus in full activity. We have subdivided them into five main groups according to their putative functions (Table 1). 

#### 2.1.1. Sugars Transporters

*M. tuberculosis* metabolism presents particularities, such as its ability to survive within the macrophage where nutrient limitations exist, when compared to other free-living bacteria, such as *M. smegmatis*, and other pathogenic bacteria, such as *Escherichia coli* and *Streptococcus pneumoniae* [33]. *M. tuberculosis*, through the glyoxylate cycle, uses lipids as the main carbon source for survival in mice. However, this fact may obscure the importance of carbohydrate transporters. The deletion of the disaccharide transporters has proven to influence infection and pathogenicity [34,35], showing that the bacillus performs metabolic transition depending on its location in the host. The bacterium has four operons that code for ABC transporters dedicated to carbohydrate transport, unlike *M. smegmatis*, an environmental species that has 19 ABC sugar transporters, highlighting the differences imposed by the different natural environments of these microorganisms [36,37]. 

One of the sugar importers identified in the *M. tuberculosis* genome is encoded by the *lpqYsugABC* operon responsible for the expression of the SBP LpqY, the TMDs SugAB and SugC, and the NBD. The putative function of this transporter is the recycling of trehalose, a particular disaccharide not present in mammals but that is formed during the production of mycolic acids [14]. Structurally, the transporter is predicted have six transmembrane helices in each of the TMDs, as predicted by the TopCons server [12]. The SugC domain shows a high similarity with the MalK protein from *E. coli*, where the presence of a regulatory domain (Table 1, represented in clear purple) is also possible. Biophysical assays on the SugC domain of this importer have shown a classical NBD behavior with optimal activity at pH 7.5 [38]. Trehalose metabolism appears to be involved in vitro in biofilm formation, which in turn is associated with the in vivo drug resistance of mycobacteria [13]. Studies have shown that trehalose analogues are able to inhibit biofilm formation and decrease bacterial growth in *M. smegmatis*. In the same study, it was also shown that this transporter is essential for anti-growth and anti-biofilm activities, hence proving its importance for the capture of trehalose and trehalose analogues [13]. Finally, the same transporter is also required for the growth of mycobacteria in C57BL/6J mouse spleen [34] and survival in primary murine macrophages [39]. 

The *uspABC* operon encodes another *M. tuberculosis* sugar transporter (UspABC) consisting of a SBP (UspC) and two TMDs (UspAB) [9]. The absence of the NDB domain in the operon suggests that this transporter might share this domain with other transporters. The TMDs are predicted to have six TM helices each. The SBP UspC contains an N-terminal extension (residues 7–29) that is responsible for anchoring the domain to the membrane. The three-dimensional structure of UspC was determined in 2016. Biochemical assays demonstrated that UspC preferentially bound amino sugars such as chitobiose, a characteristic that shows the ability of the bacillus to use scarce sources of nutrients during intracellular infection and the recycling of amino acids from its own cell wall [15] (Table 2). UspABC has been reported as an essential transporter for in vitro growth in transposon mutagenesis assays [40,41]. 

The third ABC sugar transport system in *M. tuberculosis*, UgpABCE, is formed by the SBP UgpB, the TMDs UgpA and UgpE, and the NDB UgpC [9] (Table 1). This transporter has been associated with the binding of glycerolphosphocholine (GPC) and other glycerolphosphodiesters [16] (Table 2). It appears to play an important role in the recycling of glycerophospholipid metabolites [17]. It is probable that *M. tuberculosis* uses these substrates as a source of nitrogen and carbon inside the phagosome. Genes of this transporter have been defined as non-essential during in vitro growth [46], but essential during infection [34]. Finally, the Rv2038c/39c/40c/41c transporter also binds sugars but its specific substrate is still unknown. Genes from this transporter were reported as non-essential for in vitro growth [46,47] but play an essential role during infection in vivo in mice [40]. The SBP Rv2041c was highly expressed in conditions of low pH and oxygen depletion and in phagosome infection, both in dormant and active infections. Together with other *M. tuberculosis* antigens, this protein transporter was successfully used for the serodiagnosis of patients with active tuberculosis [18] (Table 1). In addition, Rv2041c was related to high levels of TNF-α, IL-6 and IL-12p40 in bovine macrophages and induced an increase in the secretion of IFN-γ and TNF-α in lymphocytes, either in latent and active tuberculosis in mice models [48]. The structural organization of these four transporters indicates that they all belong to the type I ABC importer systems classification [9]. 

#### 2.1.2. Peptides Transporters

*M. tuberculosis* has two ABC transport systems, DppABCD and OppABCD, which are dedicated to the transport of dipeptides and oligopeptides, respectively [9] (Table 1).

The DppABCD import system has similarities with the orthologous system in *E. coli* and is formed of an SBP DppA, two TMDs (DppB-DppC) with six predicted TM helices, and an NDB (DppD). DppA shows the classic SBP type II topology, where a pocket for substrate binding is formed at the interface of the two subdomains. The three-dimensional structure of *M. tuberculosis* DppA was solved with the tetrapeptide SSVT in the ligand-binding pocket (PDB: 6E4D) [19]. Peculiarly, in the same study, the authors explored the additional possibility of DppA to bind heme and hemoglobin through the growth of a *dpp* mutant that was impaired in medium with hemin and human hemoglobin as source of iron [49]. While the *M. tuberculosis* strain with deletion in *dppD* has shown a decreasing survival in the early stages of infection, the deletion of *dppC* in the *M. bovis* bacille Calmette–Guérin (BCG) was responsible for an attenuated phenotype [50]. Finally, the DppABCD transporter is associated with the regulation of genes involved in cell wall remodeling in *M. tuberculosis*, since its deletion caused decreased survival in the lungs and spleen in mice but no change in the virulence phenotypes [34,46]. 

The transporter OppABCD has a structural organization similar to the Dpp system, consisting of an SBP (OppA), two TMDs (OppB-OppC) with six predicted TM helices each, and an NDB (OppD) (Table 1). This transporter is essential for the growth of *M. tuberculosis* in a C57BL/6J mouse spleen and infections model [34,47]. The OppABCD system was shown to bind the tripeptide glutathione and the nonapeptide bradykinin, suggesting that it can transport a wide range of substrates. In macrophages infected with *M. tuberculosis*, this transporter seems to be responsible for reducing the levels of glutathione, an ability that was lost in the *oppD* mutant. Moreover, in the *M. tuberculosis opp* deleted strain lower levels of methyl glyoxal and a decreased ability to promote apoptosis and the production of IL-6, IL-1β, and TNF-α were observed, which evidences the interference of the transporter with the innate immune response [20].

#### 2.1.3. Amino Acid Transporters

The putative ABC components related to amino acid transport identified in *M. tuberculosis* include the transporter ProXVWZ (glycine and betaine), Rv2563/64 and Rv0072/73, and the SBP GlnH (glutamine/glutamate/aspartate) [9]. Rv2563/Rv2564 and Rv0072/Rv0073 share a similar structural organization, with two NBDs and two TMDs (four helices each) that present an external periplasmic domain that spans the cell envelope (Table 1). The structural organization and function of these transporters is still uncertain. While the TMD organization is similar to that observed in the MacB family, whose members are responsible for the efflux of macrolides, division regulation, and lipoprotein or protein effectors release [51,52,53,54], the NBDs from the Rv0073 and Rv2564 share a more than 70% amino acid sequence identity with ATPases from glutamine transporter-type importers, causing some doubts about their true function [55,56]. A protein microarray study has shown that Rv2564 is immunogenic and reacts positively with serum from TB-infected patients, while showing no reactivity in control patients’ samples [21]. The transporters Rv0072/Rv0073 and Rv2563/Rv2564 are described as non-essential for *M. tuberculosis* in vitro growth, as many clinical isolates show a partial or complete deletion of Rv0072/Rv0073 [40,46,47]. 

GlnH is a periplasmic binding protein and its structure was solved in complex with glutamine, glutamate, and asparagine [22] (Table 2); it has structural feature similar to the SBPs of anion transporters. Genes up and downstream of *glnH* encode, respectively, the permease GlnX and the kinase PknG, which phosphorylates the protein GarA, either in *M. smegmatis* and *M. tuberculosis*. The phosphorylation of GarA culminates in the cancellation of the repression of the metabolism of glutamic acid [22]. Finally, the amino acid transporter ProXVWZ is similar to the glycine betaine transporters that are found in various bacteria. It is formed by the substrate binding-protein domain ProX, the TMDs ProW and ProZ, and the NBD ProV [9]. In other microorganisms, this type of transporter is responsible for the control of osmoregulation through osmoactive compounds, such as glycine betaine, proline, and others that do not show metabolic activity even in high concentrations. *M. tuberculosis* ProX binds polyphenols such as phloretin, which can be used as a carbon source [23]. Mutants of the *proXVWZ* operon showed problems in accumulating glycine betaine, deficiency in survival and growth in macrophages, and impaired the bacillus growth in a medium with a high osmolarity [57].

#### 2.1.4. Anion Transporters

*M. tuberculosis* has at least four complete ABC transporter systems for anion transport that share similar structural organization with conserved type II SBPs, TMDs with six predicted TM helices and NBDs with regulatory domains [9,58] (Table 1).

SubICysTWA is a sulfate transporter formed by the periplasmic binding protein SubI, the TMDs CysT and CysW and the NBD CysA1. This transporter is part of an important pathway for the production of L-cysteine, methionine, mycothiol and other cofactors that draw attention to them as potential targets for drug development [58,59]. Mutants of this transporter affect the growth of the bacterium in vitro and its survival in macrophages during infection [34,40,46,47]. Studies have shown that strains with mutations in *cysA1* and *subI* were unable to capture sulfur [60] and showed an increased sensitivity to several antibiotics compared to the wild-type strain where genes encoding this transporter were upregulated in depletion of sulfur [28]. The three-dimensional structure of the periplasmic binding protein SubI was solved in the presence of sulfate (PDB: 6DDN), revealing a conserved binding site (Table 2). 

ModABC is highly conserved with respect to the *E. coli* molybdate importer system and imports molybdate ions that are used as cofactor in a series of enzymes that act on the *M. tuberculosis* metabolism [61,62]. The transporter system is constituted of SBP (ModA), two TMDs (ModB) and two NBDs (ModC) (Table 1). ModC was identified as one of the antigens present in the urine of patients with active pulmonary TB it is therefore a possible biomarker for point-of-care diagnostics in combination with other MTB antigens [25]. A *M. tuberculosis* mutant lacking ModA protein has shown a lower survival rate in lungs of mice and attenuated infectious phenotype when compared to the wild-type strain but showed in vitro growth faster [46]. Moreover, the gene was non-essential during the infection of *M. tuberculosis* in macrophages [34,40]. 

Phosphate uptake in *M. tuberculosis is* organized in four operons: *pstS1pstC1pstA2pstB* (SBP-TMD-TMD-NDB), *pstS3pstC2pstA1* (SBP-TMD-TMD), *phoT* (NDB), and *pstS2* (SBP) (Table 1). It is still unclear if the uncomplete systems can be completed by components from other operons or if they assemble by combining the various components from different operons [9,63,64]. The three SBPs are present in the mycobacterial surface and they seem to be upregulated during the starvation of phosphate [26,64]. The *pstS3pstC2pstA2* expression was dependent of the two-component system SenX3-RegX3, during phosphate starvation. PstA1, PstC2, and PstS3 proteins are essential for in vivo growth and macrophage survival, and PhoT protein is essential for virulence [65]. The *pknD* gene is located downstream of the *pstS3pstC2pstA2* operon and encodes a serine/threonine kinase that plays a key role in phosphate regulation and in the activity of the transporters (Table 1). Mutations in *pknD*, induced a similar phenotype to *pstS* mutants, altering the survival ability of the bacillus in macrophages and the lungs of mice [66]. PstS1 and PstS3 have similar identical three-dimensional structures that were solved in the presence of phosphate [42,43]. Another interesting observation is that under phosphate deprivation conditions, the bacillus shows an isoniazid tolerance phenotype [67]. A high level of expression of the *pstB* in response to fluoroquinolone treatment was also observed suggesting that the expression of this gene is associated with fluoroquinolone resistance [68]. Finally, PstS proteins are highly immunogenic and hence good targets for the development of new diagnostic approaches [27] (Table 2).

#### 2.1.5. Metal Transporters 

*M. tuberculosis* uses ABC transporter components for the uptake of metals, mostly related to survival and virulence [69,70,71,72]. ABC proteins include FecB (SBP), FecB2 (SBP), Rv3041c (NDB), Rv1463 (NDB), and IrtA/IrtB, a non-canonical importer of siderophores [9,29,73,74]. The *fecB* gene encodes a substrate binding-protein homologous to *E. coli* FecB that binds iron from ferric citrate. The expression of FecB orthologous in *M. avium* was directly influenced by the iron concentration [75]. Mutants of *M. tuberculosis fecB*, have shown to be susceptible to several antibiotics, suggesting a direct relationship between FecB and intrinsic resistance to these drugs. This effect was probably due to changes in the permeability of the cell envelope caused by changes in iron metabolism [28]. The FecB2 is the second SBP identified in *M. tuberculosis* and its structure, as reported in the Protein Data Bank (PDB: 4PM4), shows a classic type III substrate binding-protein topology [76], with two subdomains αβ connected by a small α-helical hinge (Table 2). Possible functions for FecB2 include heme binding, uptake, degradation and interaction with proteins coupled to heme. The *fecB2* mutant showed reduced growth in a medium with heme as the unique iron source, in comparison with the parental strain [49]. Rv3041c consists of an NBD protein that has not yet been characterized [9]. Single nucleotide polymorphisms (SNPs) found in *rv3041c* could be responsible for phenotypes of resistance to rifampicin, ofloxacin, streptomycin and the resistance observed in MDR strains [77]. 

The *M. tuberculosis* operon *rv1461-1466* encodes a system for biogenesis of iron-sulfur cluster. Fe-S clusters are co-factors that require multiple protein systems for their biosynthesis and are related to several functions such as DNA repair, amino acid metabolism, nitrogen fixation, and others [78]. While *rv1460* encodes the operon regulator, *rv1463* encodes the NBD SufC. All the genes from this cluster are essential for *M. tuberculosis* growth [29]. During iron-limiting conditions, *M. tuberculosis* upregulates the expression of iron-related genes, such as the siderophores carboxymycobactin (cMyco) and lipophilic mycobactin [74]. 

The *irtA* and *irtB* genes encode an ABC-type transporter, with fused membrane-spanning and ATPase domains similar to those seen for Type V transporters (Figure 1). However, biochemical analyses and the three-dimensional structure of this transporter have shown that IrtA has an additional N-terminal domain, positioned in the same side of the ATPase, which works as a siderophore-interaction domain (SID) [30] (Table 2). The siderophore interaction domain from IrtA is specifically required for the transport of the lipid-bound mycobactin, but not for carboximycobactin, which also can be diffused through the inner membrane [30]. On the other hand, the siderophore domain is responsible for the uptake of carboxymycobactin from inside the cell and deliver it to the TMDs for its exportation [79]. This way, IrtA also could act as carboxymycobactin exporter working in synergy with IrtB. The mutation of *irtAB* operon in *M. tuberculosis* affects the growth in iron-deficient conditions in vitro, the efficient utilization of iron from Fe-carboxymycobactin, as well as replication of the bacillus in human macrophages [80] and in mouse lungs [40]. This transporter is essential to the bacillus survival during iron deficiency and is required for its replication in macrophages and mice [74,81]. 

#### 2.1.6. Hydrophilic Compounds

Although the transporter Rv1819c (BacA) has an overall fold similar to Type IV ABC transporters (Figure 1), with two copies of TMD, each one has six transmembrane helices that are fused to an individual NBD (Table 2). As observed in Type IV ABC transporters (exporters), the Rv1819c substrate is located in the interface between the two TMDs. Interestingly, this transporter has a large occluded water-filled cavity that spans across the whole lipid membrane. Its function was determined as an importer of hydrophilic compounds, such as vitamin B12 and bleomycin [32]. Studies have shown that a mutant bacillus of *rv1819c* loses the ability to maintain chronic infection in murine models and increases resistance to bleomycin [82]. Besides the previous functions attributed to Rv1819c, it is also speculated to play a role in the transport of peptides and drugs [31].

#### 2.1.7. Energy-Coupling Factor Transporter (ECF)

The ABC transporters type III (ECF, energy-coupling factor) are a special case of importers. These systems do not use a classic SBP, as seen for type I and II importers. The TMDs called EcfT and EcfS are structurally and functionally different (Table 1). EcfT is predicted to have between four and eight helices and makes direct contact with the two subunits of the NBDs. On the other hand, EcfS or S-component, is formed by six-helix cylinder embedded in the membrane and it is responsible for substrate binding. In some characterized ECF transporters, NBDs and the T-component interact with more than one type of S-component [82]. The operon *rv2325-rv2326* is a homologue to the ECF transporter of *E. coli* and *Lactobacillus brevis* and encodes a putative transmembrane component similar to T-component and the NBD, respectively. Recently, only the S-component was identified in the *M. tuberculosis* genome. 

### 2.2. ABC Transporters Type Exporters

*M. tuberculosis* has at least 28 identified genes encoding components of ABC transporter type exporters, from which 26 form 14 complete systems subdivided into four main functional categories: (i) recycling/transport of membrane components (two systems), (ii) electron transport chain (ETC) (one system), (iii) virulence and adaptation (two systems), and (iv) drug efflux (nine systems). The genomic organization show the genes are organized in operons that also contain related genes that, depending on the function, will help in transport. Based on secondary structure predictions, domain analyses, and the prediction of the number of transmembrane helices, these transporters show a variety of structural organizations that go beyond the classical architecture of two NBDs and two TMDs. Furthermore, both transporters and components have shown a functional plasticity in *M. tuberculosis* and related species.

#### 2.2.1. Transporters Involved in the Recycling and Transport of Membrane Components and Liposaccharides 

Two complete systems were identified in this group: RfbDE and Rv1747. RfbDE is encoded by *rv3783* and *rv3781*, respectively, which are separated by a gene encoding a putative sugar transferase (Table 3). 

The membrane-spanning (RfbD) and nucleotide-binding domains (RfbE) share homology with components of O-antigen and lipopolysaccharide exporter from different species [94,95]. RfbE and RfbD have also orthologues in *M. leprae* [96]. The inactivation of their respective orthologues in *M. smegmatis* has suggested that they are involved in the biosynthesis of cell wall components and arabinogalactan, working as a potential flippase candidate that exports the lipid intermediate containing the galactan structure prior to its arabinosylation [97]. RfbE was identified as one of the most expressed proteins in the proteomic analysis of a one-year-old dormant *M. tuberculosis* [98] bacillus and is highly abundant in multidrug-resistant and susceptible *M. tuberculosis* isolates [99]. RfbDE seems to be essential for in vitro growth [40,46] and is required for growth in C57BL/6J mouse spleen [34]. The relevance of arabinogalactan for the bacillus makes this transporter an excellent target for therapeutics. 

In the group of transporters dedicated to liposaccharides release, Rv1747 presents a quite interesting structural organization. Besides the canonical TMDs and NBDs present in exporters, it has two Fork-Head Associated domains (FHA-1 and FHA-2) in the cytoplasm side that are connected by an intrinsically disordered region of 150 amino acids [44]. The FHA are modular recognition domains with a high specificity for phospho-threonine (pThr) that participate in important cellular processes such as growth, division, differentiation, apoptosis, transcription, DNA repair, and protein degradation [100,101]. The structures of FHA-1 and FHA-2 were determined by X-ray crystallography and nuclear magnetic resonance (NMR) spectroscopy, respectively [84] (Table 2). Indeed, this transporter is phosphorylated by the *M. tuberculosis* serine/threonine protein kinase (STPK) PknF [102], a gene that is in the same operon of rv1747 [84,103,104] (Table 3). Studies have shown that the phosphorylation is essential for transporter function [105] and induces the phenomenon of the phase separation and clustering of the transporter in vitro in membrane models and live cells [84]. Although still not clear, this transporter has been associated with the translocation of bacterial cell wall components such as lipo-oligosaccharides or phosphatidyl-myo-inositol-mannosides (PIMs), antibiotic extrusion, and cell wall synthesis or remodeling [102]. Interestingly, the phosphorylated protein Rv2623 (USP) binds to Rv1747, modulating the number of PIMs in the cellular envelope [106]. Its function is essential during the infection and pathogenesis of the bacteria [84,104,106].

#### 2.2.2. Electron Transport Chain (ETC)

The operon *cydABCD* in *M. tuberculosis* is responsible for cytochrome assembly and microaerobic respiration and plays an active role in the electron transport chain [107,108] (Table 3). While the genes *cydA* and *cydB*, similarly to *E. coli*, encode subunits I and II, respectively, of the cytochrome *bd*-quinol oxidase [85], the *cydC* and *cydD* genes each encode a polypeptide consisting of the TMD and NBD that will form a heterodimer ABC transporter (type V), responsible for cytochrome bd assembly and heme transport in *M. tuberculosis* [109] and *M. smegmatis* [85]. The genes *cydABCD* are upregulated during the exposure of *M. tuberculosis* to hypoxia and nitric oxide in vitro and during the chronic phase of infection [110]. Interestingly, the disruption of *cydC* also has been associated with the persistence of the bacillus in isoniazid-treated (INH-treated) mice without affecting growth or survival in untreated mice, suggesting a connection between this transporter and resistance mechanisms in *M. tuberculosis*. Proteins from the cytochrome assembly and microaerobic respiration are considered valid targets for drug discovery [111]. 

#### 2.2.3. Virulence and Adaptation

The ABC transporter Rv0987/Rv0986 is homologous to the AttFGH transporter of *Agrobacterium tumefaciens*, which is known to be necessary for infection in plants due its participation in attachment and virulence [112,113]. In *M. tuberculosis*, *rv0986* and *rv0987* are essential for the in vitro growth of the H37Rv strain [40]. Mutants of *rv0986* showed an attenuated phenotype associated with the prevention of phagosome maturation and acidification in bone marrow macrophages from BALB/c mice [114,115] and development in the central nervous system of BALB/c mice [86]. The transporter is associated with the exportation of substrate(s) that might be critical in the infection of different cells [9]. A bioinformatics analysis of Rv0987 predicted it to have 10 to 12 transmembrane helices, with two extracellular domains forming a complex organization that is not clearly classified (Table 3) [9]. Rv0986 is closely related to FtsE (a protein that is predicted to bind ATP and hydrolyse it) from FtsXE system, an ABC transporter implicated in the cell division of *M. tuberculosis* [9,87] and the cleavage of peptideoglycans [45]. The three-dimensional structure of the extracellular domain (ECD) transmembrane FtsX component has an unusual fold that binds the N-terminal segment of RipC (Rv2190c), a peptideoglycan peptidase, inducing conformational changes in this enzyme [45] (Table 2). The interaction of *M. tuberculosis* FtsX with FtsZ (homologue of eukaryotic tubulin) was observed in vivo and ex vivo, suggesting the role of this transporter in the Z-ring formation in the divisome [87]. 

#### 2.2.4. Drug Efflux and Resistance in *M. tuberculosis*

The resistance of *M. tuberculosis* to the first and second lines of antibiotics is one of largest problems that challenges TB control strategies and therapy. The bacillus has a set of different tactics to decrease drug susceptibility, including the alteration of targets, the inactivation of the drugs, and cell envelope impermeability. The role of efflux pumps has been pointed as critical in this process. Different *M. tuberculosis* clinical isolates have shown increased levels of efflux pump gene expression, as well as the identification of critical SNPs (Single Nucleotide Polymorphisms) [4,116]. The results of these studies reveal the development of multi-drug resistant (MDR) strains that are resistant to at least two anti-TB drugs, rifampicin and isoniazid, and extensive-drug resistant (XDR) strains that are resistant to at least one of the second-line drugs [4]. 

The genome of *M. tuberculosis* has a large number of genes (2.5%) involved in drug efflux, where in the large ABC transporter family at least seven complete systems of exporters were identified, including two isolated NBDs. Five of them present a similar organization, and they are involved in the transport of macrolides and multi-drugs; these include DrrABC, Rv1686c/87c, Rv1273/Rv1272, Rv1456c/57c/58c, and Rv2686c/87c/88c. These transporters’ topology presents two TMDs with six to eight TM helices each and two NBDs homodimers (Table 3). The DrrABC transporter is implicated in doxorubicin resistance and transport of surface lipid phthiocerol dimycocerosate (PDIM) [117] and daunorubicin. In *M. smegmatis*, DrrAB has shown to confer resistance to a number of clinically relevant but structurally unrelated antibiotics. where the ATP binding was positively regulated by the compounds. Moreover, treatment with reserpine and verapamil, which are classical inhibitors of ABC transporters, was capable to impair DrrAB function [118]. DrrAB also was over-expressed in MDR isolates [88,119]. This operon is needed for growth in C57BL/6J mouse spleen [34]. Not much is described in the literature for the Rv2686c/87c/88c transporter, which is still poorly studied, but it is known to be related to the efflux of fluoroquinolone [120,121]. Although Rv1686c/87c are non-essential genes for the in vitro growth of H37Rv [46], Rv1687c is upregulated in clinical isolates [122]. Genes rv1272/73 are non-essential for the in vitro growth of H37Rv [46], yet they are required for bacillus survival in primary murine macrophages [39] and primate lungs [123]. The analysis of the XDR clinical isolates of *M. tuberculosis* revealed a common SNP in Rv1273 [124]. The genes encoding the ABC transporter Rv1456c/57c/58c were overexpressed in the presence of ethambutol, isoniazid, rifampicin, and streptomycin in *M. tuberculosis* [90]. The disruption of components from this operon results in the growth defect of H37Rv in vitro [40,46,47]. 

Structurally different from the drug exporters described previously, the Rv0194 and Rv1217c/Rv1218c systems present large TMDs of 12 predicted helices. The transporter Rv0194 has two TMDs and two NBDs encoded by a single gene (Table 3). Disruption of the *rv0194* gene promotes the in vitro growth of H37Rv [46], although proteomics studies have shown that it is a non-essential gene [47,125]. On the other hand, *rv0194* is significantly upregulated under hypoxic conditions compared to under aerobic conditions [126]. The Rv0194 transporter is highly induced in rifampicin (RIF)-resistant and RIF-susceptible isolates [91], and its role is also implicated in the efflux of ethidium bromide [127] and capreomycin/amikacin/kanamycin [128]. Rv1217c/Rv1218c transporters mediate resistance to different classes of antibiotics, such as novobiocins, pyrazolones, biarylpiperazines, bisanilinopyrimidines, pyrroles, and pyridines, and have been implicated in drug resistance in many clinical isolates [116]. The three-dimensional structure of Rv1219c, a TetR family transcriptional regulator, revealed a cavity with key aromatic residues essential for the binding of small drugs [92]. 

Finally, in the drug-efflux group, two isolated genes encode for ATPases. While Rv2477c has been associated with the efflux of kanamycin and amikacin in *M. tuberculosis* MDR and XDR clinical isolates, Rv1473 seems to be involved in macrolide transport [94].

### 2.3. Distribution of ABC Transporters across Different Species of M. tuberculosis Genus 

In order to evaluate the distribution of the ABC transporters across different species of the genus *Mycobacterium*, we have organized a panel comparing the use of all the components previously described (Figure 2). 

Amino acid sequences from *M. tuberculosis* proteins were used as a query for blastP in the Kyoto Encyclopedia of Genes and Genomes (KEGG) [129] against the data bank of relevant species, including the *M. tuberculosis* complex members, other pathogenic species, and environmental species. The analysis of the ABC transporters importer systems revealed that all the transporters identified in *M. tuberculosis* are conserved in the members of the complex. Some of these ABC importers have orthologues in all the species, such as the systems ModABC, CysWTA, OppABCD (only missing in *M. abscessus*), BacA, and ECF and the components GlnH, PstS2/PstA2, PhoT, FecB, Rv3041c, and Rv1463, suggesting that they might constitute a minimum set of importers. Nevertheless, there are other importers that are exclusive to the *M. tuberculosis* complex or just present in a few species, such as UgpABCE, which is also conserved in *M. kansasii* and *M. marinum* (two species not considered human pathogens); PstS1/PstC1/PstA1/PstB, conserved in *M. yogonense*, *M. intracellulare*, and *M. terrae*; and the transporters Rv0072/Rv0073 and Rv2563/Rv2564, also present in *M. florentinum*, *M. rhodesiae*, *M. kansasii*, *M. terrae*, and *M. fortuitum*. The analysis of the Rv0072/Rv0073 and Rv2563/Rv2564 transporters is particularly interesting since they are grouped with GlnH (ABC glutamine transporter), which is conserved in all the species. We have noticed the absence of IrtA/IrtB and FecB in *M. leprae*, *M. kansasii*, and *M. gilvum.* These systems are involved with iron/siderophores transport; hence their absence suggests that iron and siderophores are not relevant substrates for these species. Another mycobacterium that stands out is the *M. marinum*, which was the most similar to the *M. tuberculosis* complex in terms of the distribution of ABC importers. Studies of *M. marinum* have largely increased and gained interest due its genetic similarity with *M. tuberculosis* and its use as a pathogenesis model with zebra fish as the host (Figure 2). 

The ABC exporters panel has revealed that species from the *M. tuberculosis* complex group conserve all the transporter components, forming a consistent large number of exporters when compared to the other species, including the pathogenic group. This appears to suggest that, in some way, these transporters are correlated to the mechanisms of infection and pathogenesis in humans. The RfbDE, PtsXE, Rv1456c/57c/58c, Rv1473, Rv2477, Rv1272/73, Rv1747, and CydCD transporters are conserved in all species (apart from Rv1747 and CydCD that are missing in *M. leprae*), indicating that they might be part of a conserved “*core*” of ABC transporters needed for physiological functions in the genus. Two transporters stood out for their almost exclusive presence in species of the *M. tuberculosis* complex, with *M. marinum* as an exception. However, it should be noted that this species is responsible for the development of the TB in fish, which is very similar to human TB. Components related to drug efflux vary a great deal, probably reflecting the need to be exposed to xenobiotic agents and drugs. From the list, *M. leprae* has the smallest genome, with only seven out of seventeen components analyzed. Finally, the DrrABC is present in all species of the *M. tuberculosis* complex and has orthologues in the five pathogenic species *M. florentinum*, *M. marinum*, *M. ulcerans*, *M. leprae*, and *M. kansasii* (Figure 2).

### 2.4. Mce Components of Mycobacterium Tuberculosis

Although all *M. tuberculosis* ABC transporters fit structurally and functionally into the previously mentioned seven families, *M. tuberculosis* contains two (novel) ABC-like transmembrane proteins, YrbEA and YrbEB [10], and one ATPase termed MceG (mkl ATPase, Rv0655) [16] that belong to a large complex spanning the cellular envelope. YrbEA and YrbEB are encoded in four *mce* (from “Mammalian cell entry”) highly conserved operons where genes play essential roles in the entry of the microorganism into the mammalian cells and their survival within phagocytes [130,131,132] (Figure 3). 

Each *mce* operon (Mce1 to Mce4) encodes six Mce proteins (MceA-F) that span the bacillus envelope and are bound to the inner membrane as part of the ABC complex formed by the YrbEA and YrbB components. All twenty-four Mce proteins have a conserved Mce domain, with unique C-terminal domains including the cholesterol uptake porter domain, RGD motifs for probable integrin binding, and DEF motifs. The initial functions attributed to Mce proteins included an important role in the host cell invasion through cholesterol-rich regions, immuno-modulation [131] and lipid or steroid transport [123]. However, their role seems to go much beyond this, as they have been implicated in (i) pathogenesis by inhibiting alveolar macrophage activity or eliciting immune response from the host; (ii) virulence, interfering with the granuloma formation and long-term survival of mycobacteria within the host [11,132]; and, lately, (iii) altering cytokine expression through interaction with ERK1/2, promoting cell proliferation through preventing the proteasomal degradation of eEF1A1 and enabling cell adherence and entry through interactions with integrins. Therefore, the use of these proteins as diagnosis markers for mycobacterial infection has been suggested [132,133,134]. Finally, and interestingly, the *mce3* operon is absent in the non-pathogenic species *M. bovis*, *M. avium*, *M. bovis* BCG, and *M. smegmatis* [11], suggesting a specific role in virulence and survival in the human host.

## 3. Conclusions

In this review, we have compiled the most relevant data regarding ABC transporters and components from the ABC importer and exporter systems present in *M. tuberculosis.* The number and type of ABC transporters present in *M. tuberculosis* are highly conserved in species that form the MTBC, which show a similar infection phenotype, virulence, and pathogenesis. It is evident that the distribution of the importers and exporters is determined by the availability of nutrients, hosts, or types of environment they use as a habitat. MTBC species have a large number of components dedicated to phosphate uptake, opposite to the other pathogenic and environmental species, indicating that phosphate might be a relevant element for the physiology and maintenance of the microorganism in the host. 

Furthermore, many transporters have non-canonical structures, and even when they are organized in classic domains they are associated with unexpected functions. For example, the two ABC transporters IrtAB and BacA play a role as importers but have structures that are mostly similar to type IV exporters. Indeed, IrtAB has shown to export siderophore carboxymycobactin, and BacA has been related to the import of vitamin B12. 

Altogether, the features presented in this review emphasize how diverse *M. tuberculosis* ABC transporters are. They also show how much investigation still is required to fully understand the structure-function mechanisms of this large protein family in one of the most important human pathogens. 

## Figures and Tables

**Figure 1 biology-09-00443-f001:**
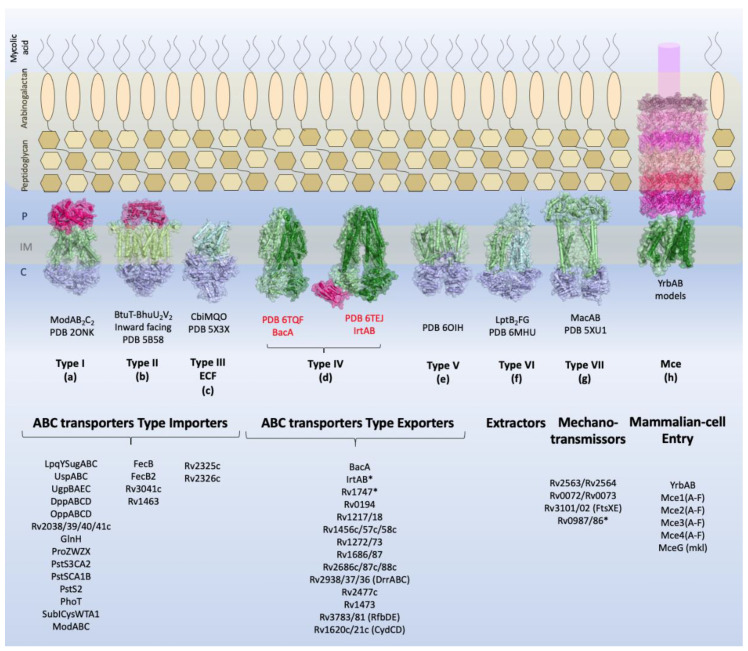
Overview of the types of ABC transporters and the components identified in *M. tuberculosis*. Schematic view of the *M. tuberculosis* cellular envelope is shown with the families of TMDABC transporters organized into importers (Types I–III), exporters (Types IV and V), extractors, and mechanotransmitters. Additionally, the structural representation of the Mce complex (Mammalian-cell entry proteins), which has two ABC-like TMDs, is shown. Representative structure of each transporter class is shown in the inner membrane: Type I, the molybdate transporter ModABC of *Archaeoglobus fulgidus* (PDB 2ONK); Type II, the vitamin transporter BtuT-BtuUV of *Burkholderia cenocepacia* (PDB 5B58); Type III, the cobalt energy-coupling factor transporter CbiMQO of *Rhodobacter capsulatus* (PDB 5 × 3X); Type IV, two ABC transporter type exporter of *M. tuberculosis* that had the three-dimensional structure resolved are shown: the ABC transporter of vitamin B12, BacA (gene Rv1819c) (PDB 6TQF), and the transporter of siderophores, IrtAB (PDB 6TEJ) (nomenclature in red); Type V, the transporter of O-antigen of *Aquifex aeolicus* VF5 (PDB 6OIH); Type VI, Cryo-EM structure of the *E. coli* LptB_2_FG transporter (PDB 6MHU); Type VII, the structure of MacAB-like efflux pump from *Streptococcus pneumoniae* (PDB 5XU1); Mce complex, prediction of the structural organization of Mce transporter systems (Mce1-4); the two ABC components, YrdAB, and the periplasmic domains, proteins MceA to MceF were modelled using I-Tasser server [8] based on *E. coli* Mla components. The structures are shown in cartoon representation with a transparent surface. The substrate-binding proteins from Type I importers and the periplasmic domains of Mce complex are shown in pink shades; the transmembrane domains are shown in dark and pale green, and the nucleotide-binding domains are shown in blue. P: periplasm, IM: inner membrane, C: cytoplasm. The ABC transporters and components of *M. tuberculosis* described in this review are listed below the representative structures. Transporters marked with asterisk (*) have structural organization that does not fit the functional or structural classification.

**Figure 2 biology-09-00443-f002:**
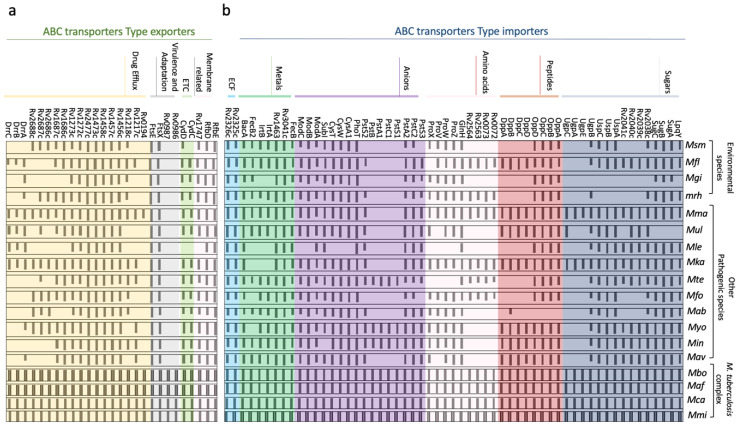
Distribution of the identified *M. tuberculosis* ABC transporters across species of *Mycobacterium* genus. (**a**) ABC export systems, (**b**) ABC (ABC) import systems. The amino acid sequences of *M. tuberculosis* components were used as a query for ortholog searching in the KEGG (Kyoto Encyclopedia of Genes and Genomes) orthologist tool [129]. The species are classified into three groups: *M. tuberculosis* complex, other pathogenic species, and environmental species. Each bar corresponds to the amino acid sequence identity (100% to 0%) of that specific protein related to the *M. tuberculosis* putative ortholog. *Msm: M. smegmatis*, *Mfl: M. florentinum*, *Mgi: M. gilvum*, *Mrh: M. rhodesiae*, *Mma: M. marinum*, *Mul: M. ulcerans*, *Mle: M. leprae*, *Mka: M. kansasii*, *Mte: M. terrae*, *Mfo: M. fortuitum. Mab: M. abscessus*, *Myo: M. yogonense*, *Min: M. intracellulare*, *Mav: M. avium*, *Mbo: M. bovis*, *Maf: Mafricanum*, *Mca: M. canettii*, *Mmi: M. microti*.

**Figure 3 biology-09-00443-f003:**
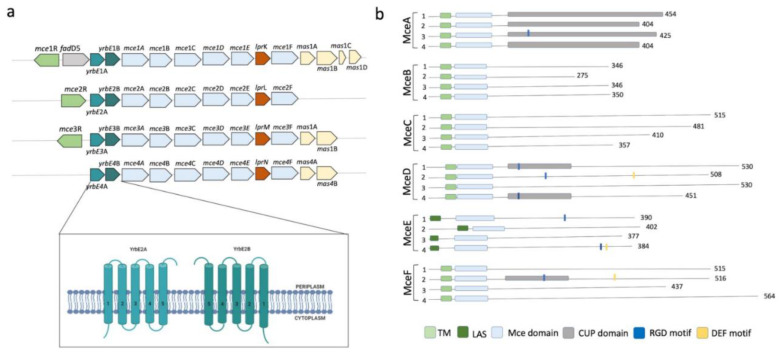
Characteristics of the Mce components of *Mycobacterium tuberculosis*. (**a**) Genomic organization of the four *mce* clusters. *mceR:* regulator (green); *fad*D5 (gray); *yrbAB*: ABC transporter integral membrane components (cyan and green, respectively); *mceA* to *mceF:* periplasmic-binding proteins (clear blue); *lprKLMN:* (red); *masABCD:* (pale yellow). The topology of the transmembrane domains is shown in detail, consisting of 10 helical transmembranes, 5 from each component. (**b**) Different domains and motifs present in the periplasmic components of MceA, MceB, MceC, MceD, and MceF. TM: transmembrane region; LAS: lipid-associated region; CUP: cholesterol uptake porter domain; RGD: motifs for putative integrin binding; DEF: probably motif for ERK2 binding.

**Table 1 biology-09-00443-t001:** Genomic context and structural organization of the ATP-Binding Cassette (ABC) importers in *Mycobacterium tuberculosis*. Importers are presented in seven groups organized by putative function. Genes that belong to the genomic context are shown as arrows. The number of helices and structural organization predicted with TOPCONS [12] are shown in the structural organization column. Substrate-binding proteins are represented in pale pink; transmembrane domains in light and dark green; nucleotide-binding domains in purple; and the regulatory domain, when present, in pale purple. SID: siderophore interaction domain.

IMPORTERS
ABC Systems/Genomic Organization	Genomic Organization	Structural Organization
**Sugars**
LpqY/SugA/SugB/SugC (Rv1235/36/37/38)	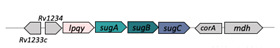	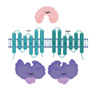
SBP-TM_1_-TM_2_-[NBDr]_(2x)_
Trehalose recycling involved in virulence and biofilm formation
[13,14]
UspC/UspB/UspA (Rv2318/17/16)	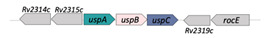	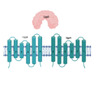
SBP-TM_1_-TM_2_-[NBDr]_(2x)_
Amino-sugars
[9,15]
UgpB/UgpA/UgpE/UgpC (Rv2833c/32c/34c/35c)	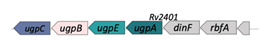	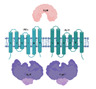
SBP-TM_1_-TM_2_-[NBDr]_(2x)_
Glycerophosphocholine
[16,17]
Rv2041c/Rv2040c/39c/38c	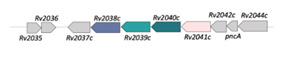	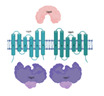
SBP-TM_1_-TM_2_-[NBDr]_(2x)_
Amino-sugar
Rv2041c—potential use for serodiagnostic and
Vaccine development
[18]
**Peptides**
DppA/DppB/DppC/DppD (Rv3666c/65c/64c/63c)	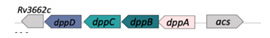	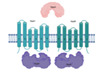
SBP-TM_1_-TM_2_-[NBDr]_(2x)_
Dipeptides
(heme and hemoglobin?)
[19]
OppA/OppB/OppC/OppD (Rv1280c/83c/81c/82c)	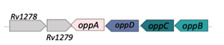	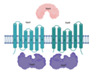
SBP-TM_1_-TM_2_-[NBDr]_(2x)_
Oligopeptides (glutatione and bradykinin)
[20]
**Amino acids**
Rv2563/Rv2564	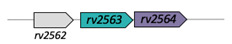	
Lipopolysaccharide export
[ECD/TM/NBDr]_(2x)_
Type VII
Rv2564 is a potential biomarker for diagnosis development
[21]
Rv0072/Rv0073	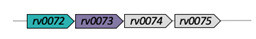	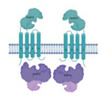
Lipopolysaccharide export
[ECD/TM/NBDr]_(2x)_
Type VII
[9]
GlnH (Rv0411c)	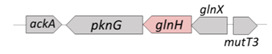	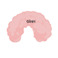
SBP
glutamine/glutamate/aspartate
[22]
ProX/ProW/ProZ/ProV (Rv3759/5756/58)	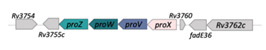	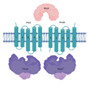
SBP-[TM/NBDr]_(2x)_
Glycine/betaine/L-proline/carnitine/
choline
[23]
**Anions**
SubI/CysT/CysW/CysA1 (Rv2400/99/98/97)	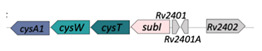	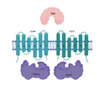
SBP-TM_1_-TM_2_-[NBD]_(2x)_
Sulfate
Members of sulfate transporter and sulfate assimilation pathway are essential and targets for drug development [24]
ModA/ModB/ModC (Rv1857/Rv1858/Rv1859)	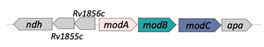	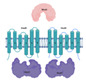
SBP-[TM/NBDr]_(2x)_
Molybdate
ModC is a good target as biomarker and potential for vaccine development [25]
PstS3/PstC2/PstA2 (Rv0928/29/30)	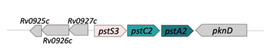	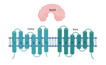
SBP-TM_1_-TM_2_
Phosphate
SBPs are potential targets for inhibition and vaccine development [26]
PstS1/PstC1/PstA1/PstB* (Rv0934/35/36/33)	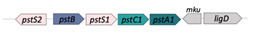	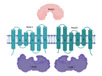
SBP-TM_1_-TM_2_-[NBD]_(2x)_
Phosphate
Potential drug targets; PstS1 is a good biomarker for diagnosis [27]
PstS2 (Rv0932)	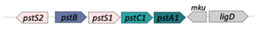	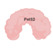
SBP
Phosphate [26]
PhoT (Rv0820)	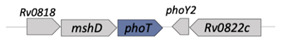	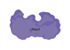
NBD
Phosphate [9]
**Metals**
FecB (Rv3040c)	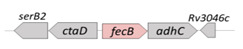	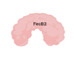
SBP
FeIII-dicitrate
Potential drug target and vaccine development
[28]
FecB2	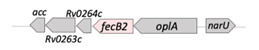	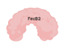
(Rv0265c)
SBP
Iron/heme [9]
Rv3041c	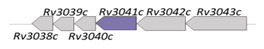	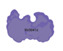
NBD
iron-hydroxamate [9]
Rv1463	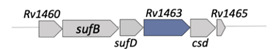	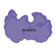
NBD
Fe-S cluster assembly [29]
SBP/TM_1_/NBD_1_-TM_2_/NBD_2_	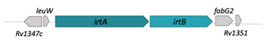	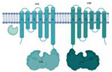
Siderophore [30]
**Hydrophilic compounds**
BacA* (Rv1819c)	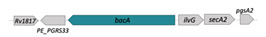	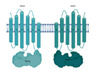
[TM/NBD]_(2x)_
Vitamin B12 [31,32]
**Energy-Coupling Factors**
Rv2325c/Rv2326c	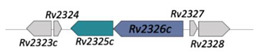	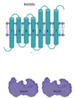

**Table 2 biology-09-00443-t002:** *M. tuberculosis* ABC transporters whose structure was solved. FHA-1 and FHA-2: Fork-Head Associated domains (1 and 2) from the Rv1747 transporter; SBP: substrate-binding protein; ECD: extracellular domain.

Transporter Component	3DStructure	PDB	Ligand	Reference
Rv2318 (UspC)SBP type II, sugar	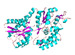	5K2Y; 5K2X	-	[15]
Rv2833c (UgpB)SBP type II, sugar	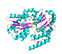	6R1B;4MF1	GPC-	[17]
Rv3666c (DppA)SBP type II, peptides	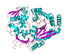	6E4D	SSVT	[19]
Rv0411c (GlnH)SBP type II, amino acids	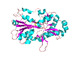	6H206HIU6H2T	AsnAspGlu	[22]
Rv2400c (SubI)SBP type II, anion	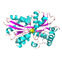	6DDN	SO_4_	-
Rv0928 (PstS3)SBP type II, anion	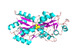	4LVQ	PO_4_	[42]
Rv0934 (PstS1)SBP type II, anion	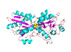	1PC3	PO_4_	[43]
Rv0263c (FecB2)SBP type III, iron	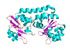	4PM4	-	-
Rv1348/Rv1349 (IrtAB), full transporter, iron	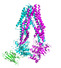	6TEJ; 6TEK	--	[30]
Rv1819c (BacA)Full transporter	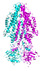	6TQF;6TQE	AMP-PNP	[32]
Rv1747_FHA-1Lipooligosaccharides, drugs	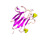	6CCD	-	[44]
Rv1747_FHA-2Lipooligosaccharides, drugs	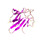	6CAH	-	[44]
Rv3101 (FtsX)ECD, division	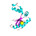	4N8N4N8O	-	[45]

**Table 3 biology-09-00443-t003:** Genomic context and structural organization of ATP- binding cassette (ABC) export systems in *Mycobacterium tuberculosis*. The ABC exporters are divided into four groups according to their function. Genes that belong to the genomic context are shown as arrows. The predicted number and structural organization of TM helices and structural organization according TOPCONS are shown in the last column, where transmembrane domains (TMDs) are colored in clear and dark green, nucleotide-binding domains (NBDs) in purple, and regulatory domains (when present) in pale purple. FHA is the fork-head associated domain and ECD is the extracellular domain.

EXPORTERS
ABC Systems/Genomic Organization	Genomic Organization	StructuralOrganization
**Recycling of Membrane compounds/Liposaccharides**
RfbD/RfbE (Rv3783/Rv3781)	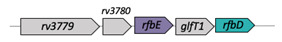	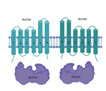
[NBDr/TM/ECD]_(2x)_
Cell wall biosynthesis,
ABC-2 subfamily of integral membrane proteins [9]
* Rv1747	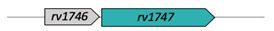	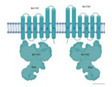
[FHA_(2x)_/NBD/TM]_(2x)_
Lipo-oligosaccharides/drug efflux [83]
**Electron transport chain (ETC)**
CydC/CydD (Rv1620c/Rv1621c)	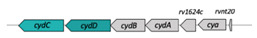	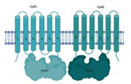
TM_1_/NBD_1_-TM_2_/NBD_2_
Cytochrome biosynthesis [84]
**Virulence, adaptation**
* Rv0987/Rv0986	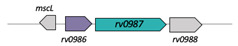	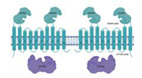
Adhesion component
[ECD_2x_/TM/NBD]_(2x)_
[85]
FtsX/FtsE (Rv3101/Rv3102)	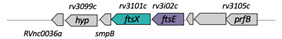	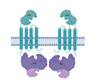
Cell division
[ECD/TM/NBD]_(2x)_
[86]
**Drug efflux**
DrrC/DrrB/DrrA (Rv2938/37/36)	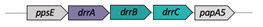	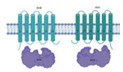
TM_1_/TM_2_-[NBD]_(2x)_
Daunorubicin/doxorubicin [87]
Rv2686c/Rv2687c/Rv2688c	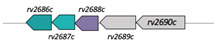	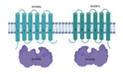
TM_1_/TM_2_-[NBD]_(2x)_
Fluoroquinolones
[88]
Rv1686c/Rv1687c	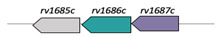	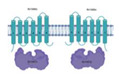
[TM/NBD]_(2x)_
Multidrug efflux [9]
Rv1273/Rv1272	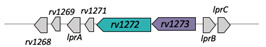	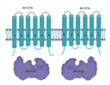
[TM/NBD]_(2x)_
MSBA subfamily/Drug efflux [87]
Rv1456c/Rv1457c/Rv1458c	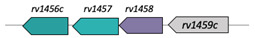	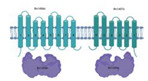
Antibiotic transport
TM_1_/TM_2_-[NBD]_(2x)_ [89]
Rv0194	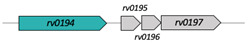	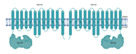
Drug efflux transport
[TM/NBD]_(2x)_ [90]
Rv1217c/Rv1218c	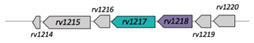	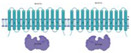
[TM/NBD]_(2x)_ [91]
Rv1473	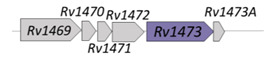	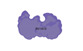
NBD
Macrolides efflux [92,93]
Rv2477c	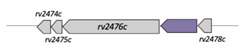	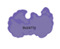
NBD
Macrolides efflux [93]

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
