# Peer review of "The ATP-Binding Cassette (ABC) Transport Systems in Mycobacterium tuberculosis: Structure, Function, and Possible Targets for Therapeutics"

_biology, 2020, doi:10.3390/biology9120443_

Round 1
Reviewer 1 Report
In the present manuscript entitled “The ATP-binding cassette (ABC) transport systems in 1 Mycobacterium tuberculosis: Structure, Function and 2 Possible Targets for Therapeutics” The author compiled the most relevant information regarding the importers and exporters present in M. tuberculosis. According to me this review is well written and provides detailed information regarding M. tuberculosis transporter system. I only find minor English grammatical issues. The author should proofread the manuscript.
Author Response
Answer to Reviewers’ comments
We thank the reviewers for the useful comments and present below all the corrections as requested point by point.
Reviewer 1
Comment. “In the present manuscript entitled “The ATP-binding cassette (ABC) transport systems in 1 Mycobacterium tuberculosis: Structure, Function and 2 Possible Targets for Therapeutics” The author compiled the most relevant information regarding the importers and exporters present in M. tuberculosis. According to me this review is well written and provides detailed information regarding M. tuberculosis transporter system. I only find minor English grammatical issues. The author should proofread the manuscript.”
Answer: we thank the comments of the reviewer. The manuscript was carefully reviewed for corrections. We also included the Summary, as requested. Please see the “biology-978616 V0_reviewed_marked.pdf”.

Reviewer 2 Report
In this article the Authors present a detailed review of the ABC transport systems identified so far in Mycobacterium tuberculosis. The manuscript is accompanied by Tables and Figures that help the reader throughout the text.
I do not have major concerns. I recommend revision of the English language and of punctuation, which is frequently misused. In addition, I noticed some mistakes that require attention:
- Page 3, legend to Figure 1, lines 91 and 92: “type VII” is indicated twice. The first one should probably be “type VI”.
- Figure 1. Please correct “Peptideglycan” to “Peptidoglycan” and “Arabinogalactin” to “Arabinogalactan”.
- Table 2 caption: please correct to “M. tuberculosis ABC transporters whose structure was solved….”.
- Lines 263-264: what do the Authors mean with “Disruption of the modA gene has shown to be benefic to in vitro growth…”? Do they mean “beneficial”? Do they mean that a modA KO strain grows better or faster in vitro?
- Line 396: please correct “electric transport chain” to “electron transport chain”.
- Line 401: please correct “exposition” to “exposure”.
- Line 429: what do the Authors mean with “Clinical isolates of different species from the M. tuberculosis”? M. tuberculosis is a species of the Genus Mycobacterium. Do the Authors mean “Different species of the M. tuberculosis complex”? Or, do they mean “Different species of the Genus Mycobacterium”? Or, do they mean “Different M. tuberculosis clinical isolates”?
- Line 471: what do the Authors mean with “different species of M. tuberculosis genus”? M. tuberculosis is a species of the Genus Mycobacterium. Do the Authors mean “Different species of the M. tuberculosis complex”? Or, do they mean “Different species of the Genus Mycobacterium”? Please see above point.
- Please correct “M. kansassii” to “M. kansasii” throughout the text.
- Please correct “M. rhodesiaea” to “M. rhodesiae” throughout the text.
- Please correct “M. microtii” to “M. microti” throughout the text.
Author Response
Reviewer 2
We thank the reviewer for the suggestions. The text was carefully corrected. Answers to specific points are presented below:
Comment 1. Page 3, legend to Figure 1, lines 91 and 92: “type VII” is indicated twice. The first one should probably be “type VI”.
Answer: type VII was corrected by type VI, as indicated.
Comment 2. Figure 1. Please correct “Peptideglycan” to “Peptidoglycan” and “Arabinogalactin” to “Arabinogalactan”.
Answer: correction made as indicated.
Comment 3. Table 2 caption: please correct to “M. tuberculosis ABC transporters whose structure was solved….”.
Answer: correction made as indicated.
Comment 4. Lines 263-264: what do the Authors mean with “Disruption of the modA gene has shown to be benefic to in vitro growth…”? Do they mean “beneficial”? Do they mean that a modA KO strain grows better or faster in vitro?
Answer: Yes. The authors showed that modA KO strain grows faster in vitro than the wild type. The sentence was written more clearly.
Comment 5. Line 396: please correct “electric transport chain” to “electron transport chain”.
Answer: correction made as indicated.
Comment 6. Line 401: please correct “exposition” to “exposure”.
Answer: correction made as indicated.
Comment 7. Line 429: what do the Authors mean with “Clinical isolates of different species from the M. tuberculosis”? M. tuberculosis is a species of the Genus Mycobacterium. Do the Authors mean “Different species of the M. tuberculosis complex”? Or, do they mean “Different species of the Genus Mycobacterium”? Or, do they mean “Different M. tuberculosis clinical isolates”?
Answer: Sorry for that mistake, the sentence was corrected: “Different M. tuberculosis clinical isolates ….”
Comment 8. Line 471: what do the Authors mean with “different species of M. tuberculosis genus”? M. tuberculosis is a species of the Genus Mycobacterium. Do the Authors mean “Different species of the M. tuberculosis complex”? Or, do they mean “Different species of the Genus Mycobacterium”? Please see above point.
Answer: We mean different species from M. tuberculosis genus (lines 482-483)
Comment 9. Please correct “M. kansassii” to “M. kansasii” throughout the text.
Answer: correction made in all the manuscript.
Comment 10. Please correct “M. rhodesiaea” to “M. rhodesiae” throughout the text.
Answer: correction made as indicated in the manuscript.
Comment 11. Please correct “M. microtii” to “M. microti” throughout the text.
Answer: correction made as indicated in the line 495.
We also included the summary as requested by the editor.
